# Validation of color Doppler ultrasound and computed tomography in the radiologic assessment of non-malignant acute splanchnic vein thrombosis

Lukas Sturm[1,2]*, Dominik Bettinger[1,2], Christoph Klinger[3], Tobias Krauss[4], Hannes Engel[4], Jan Patrick Huber[1], Arthur Schmidt[1], Karel Caca[3], Robert Thimme[1], Michael Schultheiss[1]

1 Department of Medicine II, Medical Center University of Freiburg, Faculty of Medicine, University of Freiburg, Freiburg, Germany, 2 Berta-Ottenstein-Programme, Faculty of Medicine, University of Freiburg, Freiburg, Germany, 3 Department of Medicine, RKH Hospital Ludwigsburg, Ludwigsburg, Germany, 4 Department of Radiology, Medical Center University of Freiburg, Faculty of Medicine, University of Freiburg, Freiburg, Germany

* lukas.sturm.med@uniklinik-freiburg.de

**Data Availability Statement:** The underlying data set is provided as Supporting Information 'S1 File.'

## Abstract

### Introduction

International guidelines propose color Doppler ultrasound (CDUS) and contrast-enhanced computed tomography (CT) as primary imaging techniques in the diagnosis of acute splanchnic vein thrombosis. However, their reliability in this context is poorly investigated. Therefore, the aim of our study was to validate CDUS and CT in the radiologic assessment of acute splanchnic vein thrombosis, using direct transjugular spleno-portography as gold standard.

### Materials and methods

49 patients with non-malignant acute splanchnic vein thrombosis were included in a retrospective, multicenter analysis. The thrombosis' extent in five regions of the splanchnic venous system (right and left intrahepatic portal vein, main trunk of the portal vein, splenic vein, superior mesenteric vein) and the degree of thrombosis (patent, partial thrombosis, complete thrombosis) were assessed by portography, CDUS and CT in a blinded manner. Reliability of CDUS and CT with regard to portography as gold standard was analyzed by calculating Cohen's kappa.

### Results

Results of CDUS and CT were consistent with portography in 76.6% and 78.4% of examinations, respectively. Cohen's kappa demonstrated that CDUS and CT delivered almost equally reliable results with regard to the portographic gold standard (k = 0.634 [p < 0.001] vs. k = 0.644 [p < 0.001]). In case of findings non-consistent with portography there was no

**Funding:** The article processing charge was funded by the Baden-Wuerttemberg Ministry of Science, Research and Art and the University of Freiburg in the funding programme Open Access Publishing.

**Competing interests:** The authors have declared that no competing interests exist.

clear trend to over- or underestimation of the degree of thrombosis in both CDUS (60.0% vs. 40.0%) and CT (59.5% vs. 40.5%).

## Conclusions

CDUS and CT are equally reliable tools in the radiologic assessment of non-malignant acute splanchnic vein thrombosis.

## Introduction

The clinical management of acute splanchnic vein thrombosis can be challenging as affected patients are at risk of developing severe complications such as mesenteric ischemia [1]. Therapeutic options include conservative treatment by anticoagulation or interventional treatment by transjugular lysis or implantation of a transjugular intrahepatic portosystemic shunt (TIPS). While interventional treatments offer a better chance of resolving the thrombus, they are associated with significantly higher rates of bleeding complications [2–4]. Besides the patient's clinical presentation, the grade of thrombosis and its extent in the splanchnic venous system are the most important factors for making the decision between conservative and interventional treatment, as these parameters are decisive prognostic factors [5, 6]. Hence, accurate radiologic assessment of the thrombosis is of major importance. International guidelines propose color Doppler sonography (CDUS) and contrast-enhanced computed tomography (CT) as primary imaging techniques for diagnosing acute splanchnic vein thrombosis [7, 8]. Importantly, these recommendations are based on a surprisingly small number of studies that have investigated CDUS and CT in this context, as summarized in **S1 Table**. Against this background, the aim of our study was to validate CDUS and CT in the radiologic assessment of non-malignant acute splanchnic vein thrombosis, using direct transjugular spleno-portography as gold standard.

## Materials and methods

### Patient selection and assessment of thrombosis

63 patients who received interventional treatment of acute splanchnic vein thrombosis at the University Medical Center Freiburg or the RKH Hospital Ludwigsburg, Germany, between January 2010 and December 2018 were screened. Inclusion criteria were initial diagnosis of a cirrhotic or non-cirrhotic thrombosis of the splanchnic venous system confirmed by direct transjugular spleno-portography and the presence of an additional CDUS and/or contrast-enhanced CT including a portal venous phase in the course of diagnostic work-up prior to portography. Exclusion criteria were cavernous transformation of the thrombosis, malignant thrombosis and lacking imaging documentation of the radiological examinations. After application of these exclusion criteria, 49 patients were enrolled in the study. 26 patients (53.1%) received portography, CDUS and CT, while 8 patients (16.3%) received portography and CDUS and 15 patients (30.6%) received portography and CT.

The study was a non-interventional, retrospective analysis. Imaging documentation of portography and CDUS was reviewed by two experienced interventional hepatologists (LS, MS), CT was reviewed by two experienced radiologists (TK, HE). Imaging documentation was reviewed in a blinded manner, id est investigators were not aware of the corresponding results of the other imaging techniques. Five regions of the splanchnic venous system were assessed

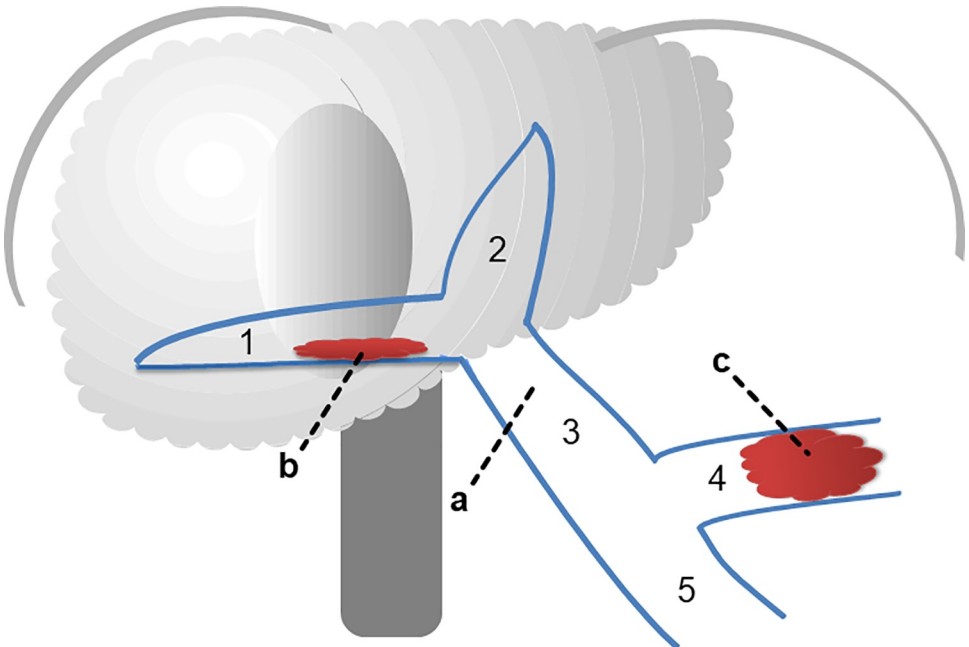

**Fig 1. Assessment of thrombosis in the splanchnic venous system.** Five regions of the splanchnic venous system were assessed: The right (1) and left (2) intrahepatic portal vein, the main trunk of the portal vein (3), the splenic vein (4) and the superior mesenteric vein (5). For each of the vessels a classification into three grades of thrombosis was made: no thrombosis (a), partial thrombosis (b) or complete thrombosis (c).

separately: Left and right main branch of the intrahepatic portal vein, main trunk of the portal vein, splenic vein and superior mesenteric vein (SMV). As CDUS examination of the SMV had not been performed routinely, the SMV could be assessed by CDUS in only 9 patients (18.4%). A classification into three degrees of thrombosis was made for each vessel: No thrombosis, partial thrombosis and complete thrombosis of the lumen; **Fig 1**. Thrombosis was defined as presence of thrombotic material in the vessel lumen with preserved (partial thrombosis) or absence (complete thrombosis) of blood flow correlate. Grading of thrombosis was conducted on a consensus basis. Since all imaging examinations were performed during clinical routine technical specificities like model of CDUS/CT and device settings were not incorporated in the analysis. The findings of direct transjugular spleno-portography were defined as gold standard, as this procedure allows direct catheter-guided application of contrast agent and visualization of the respective vessels of the splanchnic venous system. The presence of liver cirrhosis was assessed as recorded in the patients' medical files. The study was performed according to the STROBE-guidelines [9].

### Direct spleno-portography

The transjugular, transhepatic technique which has been described previously was used for direct spleno-portography in all patients [10]. In summary, an approach via the internal jugular vein was established and hepatic vein catheterization was performed using a curved cannula together with a corresponding 5 F catheter and a guidewire. Transhepatic puncture of a right portal vein branch was performed under ultrasound guidance. After successful puncture, a 5 F pigtail catheter was advanced into the splenic vein and placed close to the splenic hilum for direct portography. Routinely, 20 ml of contrast agent were injected in two seconds (10 ml/s). In case of larger collaterals leading to diversion of contrast agent before entering the portal

vein, portography war repeated in a position closer to the portal confluence. Subsequently, the catheter was repositioned for assessment of the SMV. Portography was performed in the course of interventional treatment of acute splanchnic-vein thrombosis, but prior to any interventional therapeutic measures.

### Ethics statement

The multicenter study was approved by the ethics committee of the University of Freiburg, Germany, (no. EK 85/19) and is in accordance with the Declaration of Helsinki. Due to the retrospective design of the study the requirement for informed patient consent was waived. Patient records were accessed between 1st January and 31st January 2020 to obtain the study data. All patient data were pseudonymized. The study was registered in the German Clinical Trials Register (DRKS 00019008).

### Statistical analyses

Continuous variables are expressed as mean with standard deviation, categorial variables as frequencies and percentages. Differences between patient groups were assessed by non-parametric tests such as the Chi square test, as there was no Gaussian distribution of the data. A p-value < 0.050 was considered statistically significant. We analyzed if the findings of CDUS and CT were consistent with portography for each of the five regions of the splanchnic venous system. Consistency was defined as identical degree of thrombosis in CDUS/CT and portography and is reported as frequency of consistent findings. If the grading of CDUS/CT was not consistent with portography, we analyzed if CDUS/CT showed a higher or a lower degree of thrombosis than portography. To evaluate the reliability of CDUS and CT with respect to the portographic gold standard, Cohen's kappa was calculated as a measure of inter-rater reliability. Cohen's kappa was classified according to Landis & Koch [11]; **S2 Table**. Analyses were performed with SPSS (version 24.0, IBM, NY, USA) and GraphPad Prism (version 8, GraphPad Software, San Diego, CA, USA).

## Results

### Patient characteristics

**Table 1** summarizes patient characteristics. A majority of 35 of the 49 patients included (71.4%) had non-cirrhotic acute splanchnic vein thrombosis, while splanchnic vein thrombosis was associated with liver cirrhosis in 14 patients (28.6%). All patients received therapeutic anticoagulation as soon as diagnosis of splanchnic thrombosis was established by CDUS/CT. The time span between CDUS/CT and portography was 1.95 ±2.38 days.

### Findings of portography

**Fig 2** gives an overview of the findings of portography that served as reference for CDUS and CT. Complete thrombosis occurred most frequently in the SMV (56.5% of patients) and the main portal vein (53.1% of patients). Partial thrombosis was present most commonly in the main portal vein as well as the right and left intrahepatic portal vein (38.8%, 35.4% and 28.3% of patients, respectively). No thrombosis was found most often in the splenic vein (52.2% of patients). 74.5% of patients had extensive thrombosis with three or more of the splanchnic venous system's vessels affected.

**Table 1. Patient characteristics.**

| Characteristics | Number |
|---|---|
| **Patients total** | 49 |
| **Age** [years] | 56 ±13.5 |
| **Gender** | |
| male | 29 (59.2%) |
| female | 20 (40.8%) |
| **Etiology** | |
| cirrhotic thrombosis | 14 (28.6%) |
| non-cirrhotic thrombosis | 35 (71.4%) |
| myeloproliferative disease | 8 (16.3%) |
| thrombophilia | 6 (12.2%) |
| infective/inflammatory disease | 6 (12.2%) |
| post abdominal surgery | 2 (4.1%) |
| idiopathic | 13 (26.5%) |
| **Portography + CDUS + CT** | 26 (53.1%) |
| **Portography + CDUS** | 8 (16.3%) |
| **Portography + CT** | 15 (30.6%) |
| **Time CDUS/CT to portography*** [days] | 1.95 ±2.38 |

* The time span to spleno-portography was calculated from the date of CDUS or CT depending on which imaging procedure was performed first

Abbreviations: CDUS = color Doppler ultrasound, CT = contrast-enhanced computed tomography

## Comparison of CDUS and CT against portography

We analyzed if the gradings obtained by CDUS and CT were consistent with portography that served as gold standard. Overall, CDUS and CT showed almost equal consistency with portography, as the portographic results matched with CDUS in 76.6% and with CT in 78.4%; Fig 3. The gradings of thrombosis obtained by CDUS and CT matched in 74.4% of cases, indicating that despite their similar performance the results of the two imaging techniques were not fully interchangeable.

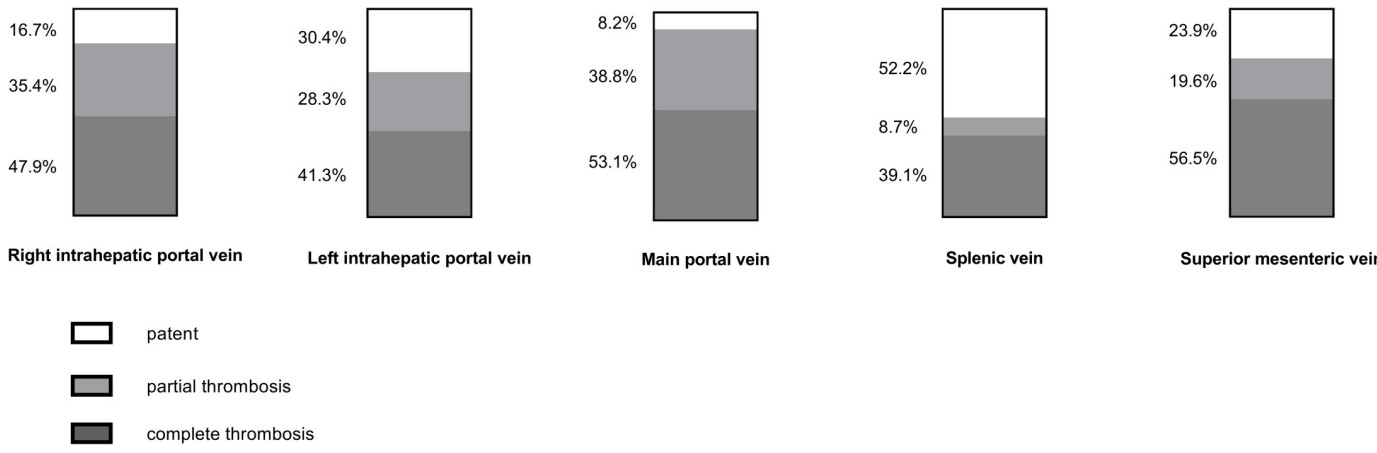

**Fig 2. Findings of portography.** The bars indicate the proportion of patients in which the five vessels of the splanchnic venous system investigated were graded as patent, partial thrombosis and complete thrombosis by the portographic gold standard that served as reference for CDUS and CT.

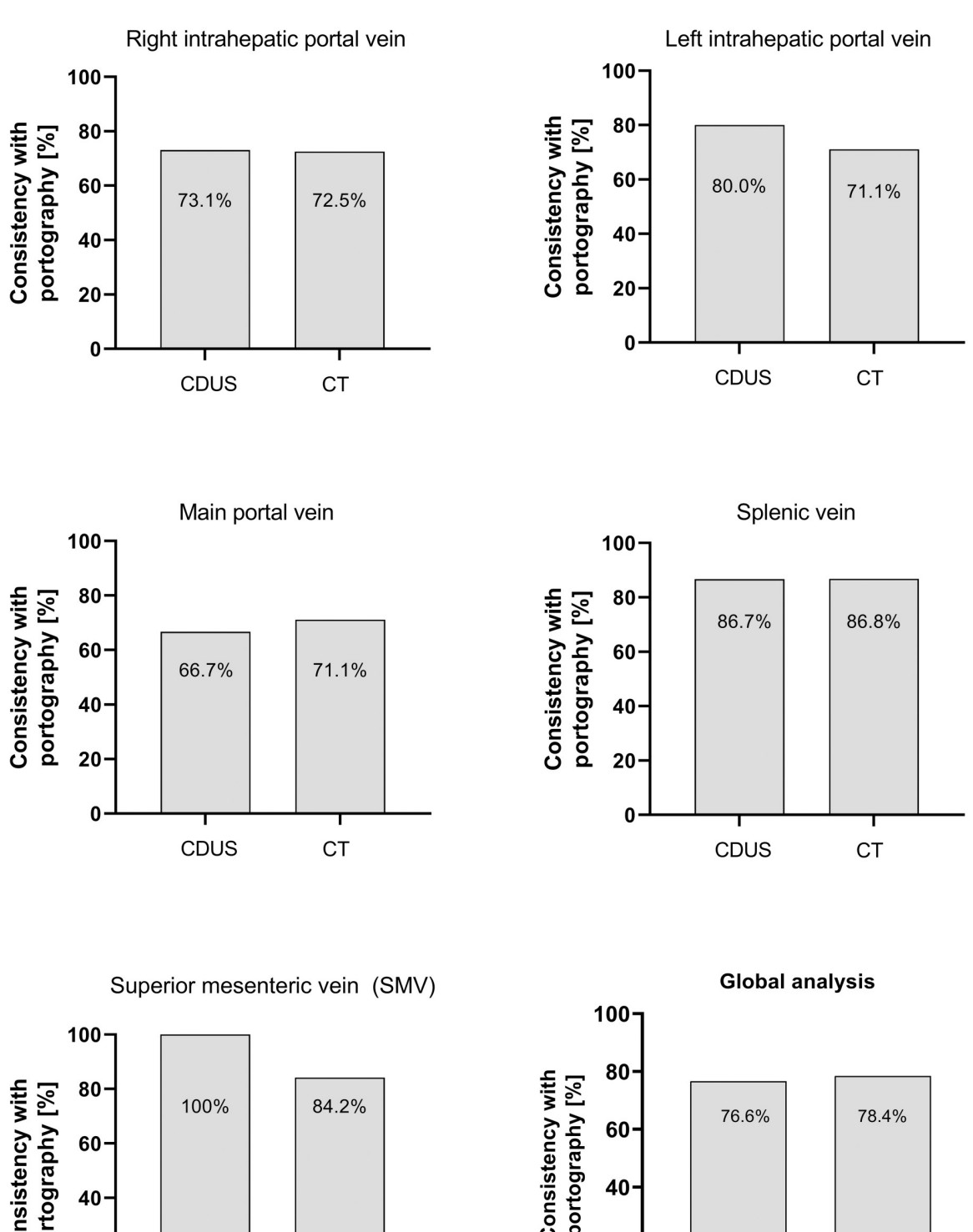

**Fig 3. Consistency of CDUS and CT with portographic findings.** The bars indicate the proportion of examinations in which findings of CDUS and CT were consistent with the portographic gold standard, id est grading of thrombosis (patent vessel, partial thrombosis, complete thrombosis) was identical in CDUS/CT and portography.

**Table 2. Cohen's kappa of CDUS and CT with regard to portography as gold standard.**

|  | Cohen's kappa | Cohen's kappa |
|---|---|---|
|  | CDUS | CT |
| **Overall** | 0.634 [p < 0.001] | 0.644 [p < 0.001] |
| **Main PV** | 0.408 [p = 0.005] | 0.593 [p < 0.001] |
| **Right intrahepatic PV** | 0.542 [p < 0.001] | 0.538 [p < 0.001] |
| **Left intrahepatic PV** | 0.688 [p < 0.001] | 0.545 [p < 0.001] |
| **Splenic Vein** | 0.681 [p = 0.001] | 0.769 [p < 0.001] |
| **SMV** | 0.999 [p < 0.001]* | 0.681 [p < 0.001] |

* The SMV was examined by CDUS in only nine patients

Abbreviations: CDUS = color Doppler ultrasound, CT = contrast-enhanced computed tomography, PV = portal vein, SMV = superior mesenteric vein

Detailed analyses of the regions of the splanchnic venous system revealed very similar rates of consistency with portography in case of the right intrahepatic portal vein (73.1% vs. 72.5%) and the splenic vein (86.7% vs. 86.8%). In the assessment of the main portal vein CDUS displayed slightly lower consistency with portography compared to CT (66.7% vs. 71.1%), while in case of the left intrahepatic portal vein CDUS was even superior to CT (80.0% vs. 71.1%). In case of the SMV consistency of CDUS with portography was 100% compared to 84.2% consistency of CT. However, as mentioned before, the SMV was assessed by CDUS in only nine patients (18.4%).

In a sub-analysis of the 26 patients in which both CDUS and CT were available in addition to portography, CDUS and CT displayed lower consistency with portography. Still, both imaging techniques performed almost equally well (66.9% vs. 68.5% consistency).

## Reliability of CDUS and CT with regard to portography

To assess the reliability of results of CDUS and CT with regard to portography as gold standard, Cohen's kappa was calculated. This analysis revealed that CDUS and CT both showed substantial agreement with portography. Cohen's kappa was k = 0.634 (p < 0.001) for CDUS and k = 0.644 (p < 0.001) for CT, indicating that the two imaging techniques displayed almost identical reliability. Subanalyses revealed that both CDUS and CT offered significant, at least moderate agreement with portography in all of the five regions of the splanchnic venous system investigated, as summarized in **Table 2**.

## Non-consistent findings of CDUS/CT and portography

**Fig 4** shows an example of divergent grading of thrombosis by CDUS in comparison to CT and portography. Systematic analysis of those cases in which the grading by CDUS/CT was not consistent with portography revealed that there was no clear trend to overestimation (CDUS 60%, CT 59.5% of cases) or underestimation (CDUS 40%, CT 40.5%) of the grade of thrombosis. Importantly, failure to detect thrombosis occurred rarely in both CDUS and CT: Complete or partial thrombosis was erroneously graded as patent in only three cases by CDUS and in five cases by CT. We assessed if the time span between initial detection of splanchnic thrombosis by CDUS/CT and portography influenced the performance of the imaging techniques. However, consistency with portographic findings was not significantly different between patients with a time span of less than 48 hours and patients with 48 hours or more between CDUS/CT and portography (p ≥ 0.296).

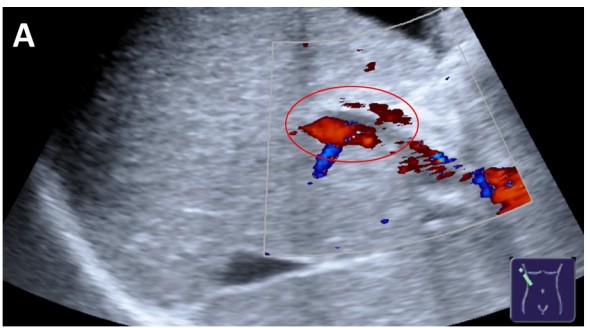
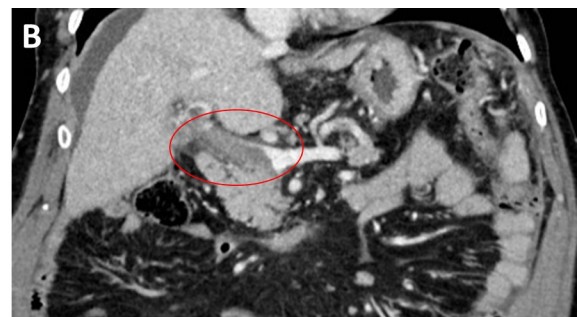
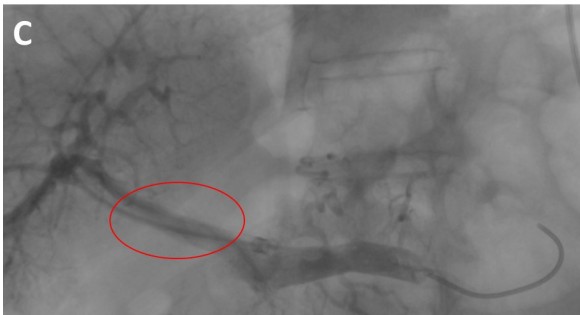

**Fig 4. Example of divergent findings in CDUS and CT.** While the main trunk of the portal vein was graded as 'partial thrombosis' by CDUS (A), since thrombotic material could be identified in the vessel lumen but a preserved portal venous flow signal was detectable, it was graded as 'complete thrombosis' by CT (B) and portography (C), as both showed thrombotic occlusion of the vessel lumen without contrast agent flow.

## Discussion

Patients with acute splanchnic vein thrombosis are at risk of developing life-threatening complications [1]. Hence, adequate treatment of the thrombosis is of great importance. The guidelines of the American Association for the Study of Liver Diseases (AASLD) and the European Association for the Study of Liver Diseases (EASL) do not give distinct criteria regarding the decision between conservative treatment and interventional treatment, but recommend to make the decision on an individual basis [7, 8]. In this context, the radiologic assessment of the thrombosis' grade and extent in the splanchnic venous system is essential, as these parameters are decisive prognostic factors and find consideration in specialist recommendations, such as the Baveno consensus report, accordingly [5, 6]. In compliance with international guidelines, CDUS and contrast-enhanced CT are the primary imaging techniques employed in the diagnosis of acute splanchnic vein thrombosis [7, 8]. Surprisingly, CDUS and CT are only very sparsely investigated in this context. As this may complicate judging the reliability of imaging results in clinical practice significantly, we set out to validate CDUS and CT in the radiologic assessment of non-malignant acute splanchnic vein thrombosis, using direct transjugular spleno-portography as gold standard. Our blinded evaluation of the imaging modalities revealed that CDUS and CT both showed comparable consistency with the grading of thrombosis obtained by the portography (76.6% vs. 78.4%). Cohen's kappa demonstrated that both CDUS and CT delivered reliable results, performing almost equally well (k = 0.634 [p < 0.001] vs. k = 0.644 [p < 0.001]). However, it has to be pointed out that CDUS and CT failed to correctly assess the grade of thrombosis in 23.4% and 21.6% of examinations, respectively, which displays that both imaging techniques involved a significant rate of erroneous gradings. Importantly, these related almost solely to the differentiation between partial and complete thrombosis and not the general presence of thrombosis.

To the best of our knowledge, there are only two other studies that have investigated CDUS and CT in the context of acute splanchnic vein thrombosis head-to-head [12, 13]. Both studies reported comparable results of CDUS and CT, but focused on the mere detection of thrombosis and did not differentiate grade and extent of thrombosis. In addition, they investigated very small patient collectives (five and three patients) and date back to 1985 and 1992. Tessler et al. investigated the main portal vein by CDUS in 75 patients against angiography/surgery as gold standard [14]. Importantly, only nine of these patients had portal vein thrombosis since the study design aimed to determine CDUS' sensitivity and specificity and did not include a more detailed analysis of the grade and extent of thrombosis. Bach et al. evaluated CDUS and computed tomography with arterial spleno-portography (CTAP) in 41 patients with splanchnic thrombosis [15]. The authors assessed the right and left intrahepatic branches and the main trunk of the portal vein and also graded thrombosis. However, patients in this study had malignant splanchnic thrombosis, thus representing a different entity of thrombosis. In addition, the authors compared CDUS to CTAP which involves contrast agent application via the mesenteric arteries, making its results only limitedly comparable to those of contrast-enhanced CT with application of contrast agent via a peripheral vein [16]. Rossi et al. performed two studies in which they compared contrast-enhanced ultrasound to CDUS in 79 patients and to contrast-enhanced CT in 50 patients with splanchnic vein thrombosis [17, 18]. Again, the included patients had malignant thrombosis and the studies focused on the detection of thrombosis and its characterization as malignant or benign. In summary, the present study is the first to directly compare CDUS and CT in the radiologic assessment of non-malignant acute splanchnic vein thrombosis, demonstrating the equal value of both imaging techniques in this context.

Our study has several limitations that need to be addressed. As acute splanchnic vein thrombosis is a very rare condition with an incidence of < 1/100.000 per year, we chose a retrospective study design, which allowed us to include 49 patients in our analysis [19]. This patient collective size is comparable with previous imaging studies on splanchnic thrombosis [12–18]. Still, it is important to keep the limited number of patients in mind when interpreting the significance our findings. Since CDUS and portography are dynamical examinations, it has to be mentioned that a post hoc analysis of their findings is inferior to an analysis during the procedure. To minimize bias regarding this aspect, portography, CDUS and CT were evaluated by two experienced investigators on a consensus basis according to set criteria and in a blinded manner. Importantly, our results may not be fully transferable to all clinical settings, as the investigator's experience may influence imaging results significantly, especially in the case of CDUS. Another limitation of our study design is that imaging examinations were performed during clinical routine over a time period of nine years and did not follow a set study protocol. Firstly, this implies a variability regarding technical aspects like model or device settings, which may impede comparability of findings. However, both participating centers were equipped with modern CDUS and CT devices of a comparable technical standard, as listed in **S3 Table**. Secondly, the SMV was examined by CDUS in clinical routine in only nine patients. As CDUS showed perfect consistency with portographic findings in evaluation of the SMV, this may be considered a source of bias. However, even when excluding the SMV from the analyses CDUS and CT still displayed comparable reliability with respect to portographic findings (CDUS k = 0.608 [p < 0.001] vs. CT k = 0.632 [p < 0.001]). Another issue that has to be discussed is effects of therapeutic anticoagulation. As all patients received anticoagulation as soon as the diagnosis of acute splanchnic vein thrombosis was established, one could assume bias through therapeutic effects of anticoagulation in imaging examinations performed henceforth. Importantly, the time span between CDUS/CT and angiography was only 1.95 ±2.38 days in our patient cohort. In addition, all patients received CDUS and/or CT prior to

portography. In case of significant bias through therapeutic effects of anticoagulation one would expect an unequivocal trend to overestimation of the grade of thrombosis in CDUS and CT, which was not true, though.

## Conclusions

Keeping its limitations in mind the results of our study demonstrate that CDUS and CT are both equally reliable in assessing the grade and extent of acute splanchnic vein thrombosis. The non-inferiority of CDUS promotes its use as an easily available, radiation-free and economical tool in this setting. Naturally, our results require validation in further studies of larger patient collectives.

## Supporting information

**S1 Table. Overview of studies investigating imaging techniques in the diagnosis of acute splanchnic vein thrombosis.**
(DOCX)

**S2 Table. Classification of Cohen's kappa.**
(DOCX)

**S3 Table. Ultrasound and CT devices utilized at the study centers.**
(DOCX)

**S1 File. Underlying data set.**
(XLSX)

## Author Contributions

**Conceptualization:** Lukas Sturm, Dominik Bettinger, Michael Schultheiss.

**Data curation:** Lukas Sturm, Dominik Bettinger, Christoph Klinger, Michael Schultheiss.

**Formal analysis:** Lukas Sturm, Dominik Bettinger, Tobias Krauss, Hannes Engel, Michael Schultheiss.

**Investigation:** Lukas Sturm, Dominik Bettinger, Tobias Krauss, Hannes Engel, Michael Schultheiss.

**Methodology:** Lukas Sturm, Dominik Bettinger, Michael Schultheiss.

**Project administration:** Lukas Sturm, Dominik Bettinger, Michael Schultheiss.

**Supervision:** Robert Thimme, Michael Schultheiss.

**Validation:** Dominik Bettinger, Michael Schultheiss.

**Visualization:** Lukas Sturm, Dominik Bettinger, Michael Schultheiss.

**Writing – original draft:** Lukas Sturm, Dominik Bettinger, Michael Schultheiss.

**Writing – review & editing:** Lukas Sturm, Dominik Bettinger, Christoph Klinger, Tobias Krauss, Jan Patrick Huber, Arthur Schmidt, Karel Caca, Robert Thimme, Michael Schultheiss.

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
