## [Decision Letter · Decision Letter 0]

31 May 2021

PONE-D-20-25875

Validation of color Doppler ultrasound and computed tomography in the radiologic assessment of non-malignant acute splanchnic vein thrombosis

PLOS ONE

Dear Dr. Sturm,

Thank you for submitting your manuscript to PLOS ONE. After careful consideration, we feel that it has merit but does not fully meet PLOS ONE’s publication criteria as it currently stands. Therefore, we invite you to submit a revised version of the manuscript that addresses the points raised during the review process.

We look forward to receiving your revised manuscript.

Kind regards,

Roza Chaireti

Academic Editor

PLOS ONE

Journal Requirements:

2. In your ethics statement in the Methods section and in the online submission form, please provide additional information about the data used in your retrospective study. Specifically, please ensure that you have discussed whether all data were fully anonymized before you accessed them and/or whether the IRB or ethics committee waived the requirement for informed consent. If patients provided informed written consent to have data from their medical records used in research, please include this information.

3. Please include the date(s) on which you accessed the databases or records to obtain the data used in your study.

4. In your Data Availability statement, you have not specified where ALL the minimal data set underlying the results described in your manuscript can be found. PLOS defines a study's minimal data set as the underlying data used to reach the conclusions drawn in the manuscript and any additional data required to replicate the reported study findings in their entirety. All PLOS journals require that the minimal data set be made fully available. For more information about our data policy, please see http://journals.plos.org/plosone/s/data-availability.

[LS andDB aresupported by the Berta-Ottenstein-Programme, Faculty of Medicine, University of Freiburg.]

 [The authors received no specific funding for this work.]

Reviewers' comments:

Reviewer's Responses to Questions

**Comments to the Author**

1. Is the manuscript technically sound, and do the data support the conclusions?

Reviewer #1: Yes

Reviewer #2: Partly

Reviewer #3: Partly

2. Has the statistical analysis been performed appropriately and rigorously? 

Reviewer #1: Yes

Reviewer #2: I Don't Know

Reviewer #3: Yes

3. Have the authors made all data underlying the findings in their manuscript fully available?

Reviewer #1: Yes

Reviewer #2: Yes

Reviewer #3: No

4. Is the manuscript presented in an intelligible fashion and written in standard English?

Reviewer #1: Yes

Reviewer #2: Yes

Reviewer #3: No

5. Review Comments to the Author

Reviewer #1: I apologize for the delay in advising you on the progress on the paper that you submitted to the PLOS ONE.

This is well written and illustrated paper. I suggest you make minor revision.

1.Is CT more reliable than CDUS if the evaluator is not an experienced interventional hepatologists/radiologists?

2.Were the CT and CDUS purchased at about the same time and with the same high performance?　 Is the CT the old one and the US the newest one?

Reviewer #2: The authors compared CDUS or CT to direct transjugular spleno-portography about detectable ability for acute splanchnic vein thrombosis. Their results is that CDUS and CT were consistent with portography in 76.6% and 78.4% compared to direct transjugular spleno-portography. They claimed that CDUS and CT are equally reliable tools in the radiologic assessment of non-malignant acute splanchnic vein thrombosis.

I think the work is well written.

However, I think the authors should consider several problems.

Major

#1. The number of enrollment was only 49 patients. I think this is too small.

#2. In clinical, CDUS and enhanced-CT are useful and often performed for diagnosis of portal or splanchnic vein thrombosis. We often perform direct transjugular spleno-portography after being diagnosed in order to intervention therapy.

So, this conclusion seems to be low impact.

#3. Concerns to patient characters, 35 patients have non-cirrhotic thrombosis. What were their detail cause ? What number was due to surgery?

Minor

#1. Images of CDUS and CT as figures should be added.

Reviewer #3: The authors present a retrospective cohort study comparing portography versus CT or CDUS for the diagnosis of splanchnic vein thromboses. Overall, the study adds new data in an area with very little information. In general, the study is well done but it could use some improvement.

Please consider a subanalysis comparing the 26 patients who had all 3 imaging modalities. Compare CT and US versus portography and also US vs CT. This also may be of value as CT and US are used interchangeably when they may be not.

It is possible that among those patients in whom there was a discrepancy between portography and the other imaging modalities the time lapse between imaging modalities might have been longer (i.e. the CDUS or CT were done first – failing to detect a thrombus, that subsequently developed and was later found by portography). Please clarify this.

Please make sure that the study adheres to STROBE guidelines.

Figure 2 could use a single panel with 5 100%-stacked columns, one per vessel. Would make it much clearer.

The manuscript needs a thorough revision of the English language. A few examples are shown below but there are many more.

Page 4. Should read “mesenteric”

Page 6. Should read “confluence”

Page 9: please rephrase the following: “In the evaluation of CDUS results complete and partial thrombosis was erroneously graded as patent in only one and two cases of the entirety of splanchnic venous regions assessed, respectively. In the evaluation of CT misjudgment of complete and partial thrombosis as patent occurred in only two and three cases, respectively”. It is not clear in its current form

Page 13. The use of the word “subsuming” is not correct

6. PLOS authors have the option to publish the peer review history of their article (what does this mean?). If published, this will include your full peer review and any attached files.

Reviewer #1: No

Reviewer #2: No

Reviewer #3: No

---

## [Author Response · Author response to Decision Letter 0]

16 Jul 2021

A point-by-point reply to the reviewers comments is included in the Cover Letter of the revision ('Rebutal Letter')

---

## [Editor Report · Decision Letter 1]

6 Dec 2021

Validation of color Doppler ultrasound and computed tomography in the radiologic assessment of non-malignant acute splanchnic vein thrombosis

PONE-D-20-25875R1

Dear Dr. Sturm,

We’re pleased to inform you that your manuscript has been judged scientifically suitable for publication and will be formally accepted for publication once it meets all outstanding technical requirements.

Kind regards,

Roza Chaireti

Academic Editor

PLOS ONE

---

## [Editor Report · Acceptance letter]

10 Dec 2021

PONE-D-20-25875R1 

Validation of color Doppler ultrasound and computed tomography in the radiologic assessment of non-malignant acute splanchnic vein thrombosis 

Dear Dr. Sturm:

I'm pleased to inform you that your manuscript has been deemed suitable for publication in PLOS ONE. Congratulations! Your manuscript is now with our production department. 

Kind regards, 

on behalf of

Dr. Roza Chaireti 

Academic Editor

PLOS ONE